# Conditions for Achieving Postoperative Pelvic Incidence-Lumbar Lordosis < 10° in Circumferential Minimally Invasive Surgery for Adult Spinal Deformity

**DOI:** 10.3390/jcm11061586

**Published:** 2022-03-13

**Authors:** Masayuki Ishihara, Shinichirou Taniguchi, Takashi Adachi, Yoichi Tani, Masaaki Paku, Muneharu Ando, Takanori Saito

**Affiliations:** Department of Orthopedic Surgery, Kansai Medical University, 2-3-1 Shinmachi, Hirakata City 573-1191, Japan; tanigucs@takii.kmu.ac.jp (S.T.); adachita@hirakata.kmu.ac.jp (T.A.); taniyoic@gmail.com (Y.T.); kmu.orthopaedics.pak@gmail.com (M.P.); mando@gaia.eonet.ne.jp (M.A.); saitot@takii.kmu.ac.jp (T.S.)

**Keywords:** adult spinal deformity, lateral lumbar interbody fusion, percutaneous pedicle screw, circumferential minimally invasive surgery, lumbosacral fusion

## Abstract

This retrospective study aimed to evaluate the clinical outcomes of circumferential minimally invasive surgery (CMIS) using lateral lumbar interbody fusion (LLIF) and percutaneous pedicle screw (PPS) in adult spinal deformity (ASD) patients, and to clarify the conditions for achieving postoperative pelvic incidence-lumbar lordosis (PI-LL) < 10°. Demographics and other parameters of ASD patients who underwent CMIS and who were divided into groups G (achieved postoperative PI-LL < 10°) and P (PI-LL ≥ 10°) were compared. Of the 145 included ASD patients who underwent CMIS, the average fused level, bleeding volume, operative time, and number of intervertebral discs that underwent LLIF were 10.3 ± 0.5 segments, 723 ± 375 mL, 366 ± 70 min, and 4.0 segments, respectively. The rod material was titanium alloy in all the cases. The PI-LL significantly improved from 37.3 ± 17.9° to 1.2 ± 12.2° postoperatively. Pre- and postoperative PI, postoperative LL, preoperative PI-LL, PI-LL after LLIF, and postoperative PI-LL were significantly larger in group P. PI-LL after LLIF was identified as a significant risk factor of postoperative PI-LL < 10° by logistic regression, and the cut-off value on receiver operating characteristic curve analysis was 20°. Sufficient correction was achieved by CMIS. If PI-LL after LLIF was ≤20°, it was corrected to the ideal alignment by the PPS procedure.

## 1. Introduction

Sagittal and coronal imbalance due to adult spinal deformity (ASD) causes pain and dysfunction in daily life [1,2]. Correction surgery for ASD results in significant improvement in health-related quality of life (HRQOL) [3,4,5,6]. Schwab et al. reported that the Oswestry Disability Index (ODI) was significantly lower in patients with sagittal alignment achieving postoperative pelvic tilt [PT] < 20°, sagittal vertical axis [SVA] < 100 mm, and postoperative pelvic incidence-lumbar lordosis (PI-LL) < 10° [7]. However, high complication and re-surgery rates have been reported to be associated with conventional open surgery [8,9,10,11]. Some elderly patients may not be able to undergo corrective surgery due to the invasiveness and incidence of complications of this conventional surgery. Introduction of a minimally invasive technique would therefore allow more patients with spinal deformities to undergo corrective surgery. In 2006, Ozgur et al. reported a novel approach called extreme lateral interbody fusion (XLIF^®^: NuVasive, Inc., San Diego, CA, USA) that enabled minimally invasive intervertebral release and interbody fusion [12]. In addition, circumferential minimally invasive surgery (CMIS) in ASD with percutaneous pedicle screw (PPS) is also gaining popularity [13,14]. However, there have been some reports on the insufficient corrective capability of CMIS for ASD [14,15]. On the other hand, there have also been few reports of clear indications for CMIS in patients with ASD. Schwab et al. reported differences in the Oswestry Disability Index between patients with SVA < 50 mm and SVA > 50 mm or between patients with PI-LL < 9° and PI-LL > 9° [7]. However, we only focused on PI-LL because only LL can be adjusted directly by surgery. Since posterior column osteotomy is not performed in CMIS, insufficient correction rather than overcorrection might occur [14,15]. Therefore, in the formula of PI − 10 < LL < PI + 10, we considered that the minimum target value, PI-LL < 10°, is an essential condition for target alignment. In this study, we evaluated the clinical outcomes of CMIS using lateral lumbar interbody fusion (LLIF) and PPS in ASD and clarified the conditions for achieving optimal postoperative alignment with postoperative PI-LL < 10°.

## 2. Materials and Methods

### 2.1. Patient Selection

Four surgeons with experience in minimally invasive spine surgery participated in the study. The inclusion criteria were age > 45 years, SVA > 50 mm, PT > 20°, pelvic incidence (PI)-lumbar lordosis (LL) > 10°, fused middle or lower thoracic to ilium, a minimum of two years of follow-up, and availability of standing full-length lateral and anteroposterior (AP) radiographs of the spine at preoperative baseline and at the final follow-up. Patients with ASD who underwent CMIS combined with LLIF and posterior spinopelvic fixation with a PPS system at our institution between January 2017 and November 2019 with a minimum follow-up period of 24 months were included. Patients who underwent surgeries from the upper thoracic to the ilium and had incomplete or inadequate radiographs for complete analysis were excluded. In addition, patients with multi-level iatrogenic kyphosis and those with multi-level posterior bone fusion that did not meet the CMIS indication criteria at our institution were excluded (Table 1). The outcomes examined were the average number of fixed vertebral bodies, the average number of intervertebral levels that underwent LLIF, the number of patients who underwent lateral access corpectomy (LAC), rod diameter and rod material, upper instrumented vertebra (UIV), intraoperative blood loss and operative time, various pre- and postoperative spinopelvic parameters, and complications. Patients were further divided into group G (those who had postoperative PI-LL < 10°) and group P (those who did not meet the postoperative PI-LL < 10° criterion). Spinopelvic parameters were compared between the two groups, and predictors of postoperative PI-LL < 10° were investigated by logistic regression analysis.

### 2.2. Surgical Method

LLIF was first performed from L1/2 (in some cases T11/12 or T12/L1) to L4/5, and one week later, posterior fixation with PPS and mini-open transforaminal lumbar interbody fusion (TLIF) were performed at L5/S1. Fixation range was from the middle or lower thoracic spine to the pelvis in all cases. We used percutaneous instrumentation and polyaxial screws at all vertebral levels in all patients and the persuader system for percutaneous instrumentation in the upper instrumented vertebra (UIV) to the pelvis. Lordotic titanium (10°) cages were used in all cases. The cage was filled with autologous iliac bone and a hydroxyapatite/collagen composite (Refit^®^; HOYA Technosurgical Co., Tokyo, Japan).

### 2.3. Radiological Evaluation

Standing full-length lateral radiograph of the spine was recorded at the preoperative baseline and final follow-up. The following spinopelvic parameters were investigated using the current standard method: PI, LL, PT, thoracic kyphosis (TK), and SVA (Figure 1). Proximal junctional kyphosis (PJK) is defined as the postoperative proximal junctional angle (PJA) between the caudal endplate of the UIV and the cephalad endplate of the UIV+ 2 ≥ 20° and at least 20° greater than the preoperative measurements [16]. Coronal imbalance was defined as ≥30 mm between the central sacral line and the mid-C7 vertebral body.

### 2.4. Statistical Analysis

Radiographic and clinical parameters were analyzed using the Mann–Whitney U test or Wilcoxon signed-rank test for continuous data and the Chi-square test for categorical data. Furthermore, logistic regression analysis and receiver operating characteristic (ROC) analysis were performed based on the results of the univariate analysis.

Statistical significance was set at a *p* value < 0.05. All analyses were performed using JMP software (SAS Institute Inc., Cary, NC, USA).

## 3. Results

### 3.1. Demographics

Based on the inclusion criteria, 156 patients were identified, and 11 patients were excluded (two underwent surgeries from the upper thoracic vertebrae to the ilium and nine did not have sufficient radiological data), leaving 145 patients available for analysis. Demographic data are shown in Table 2. The mean surgical age was 73.3 years (48–83 years), and the mean follow-up period was 40.7 months (30–54 months). Female patients accounted for 80.7% of all cases. Average blood loss in the first and second surgery was 104 ± 139.3 mL and 498.2 ± 305.7 mL, respectively. The average operative time in the first and second surgery was 109.6 ± 37.5 min and 233.3 ± 50.9 min, respectively. In comparison, the amount of blood loss and operative time were significantly less in anterior surgery than that in posterior surgery (*p* < 0.001). The average number of intervertebral levels that underwent LLIF was 4.0 ± 0.5 (3–6). We also performed LLIF at T11/12 or T12/L1 in patients with disc wedging or kyphotic deformity at the thoracolumbar junction. UIV was at T9/T10 in most cases, but in some patients with adult idiopathic scoliosis, the UIV was at T7. There were eight patients who underwent LAC, and the vertebrae that underwent LAC were T12, L1, and L2 in two cases and L3 and L4 in one case. The rod material used was titanium alloy (TA) in all cases, and the diameters and number of rods used were 5.5 mm and 2 rods in 54 cases, 6.0 mm and 2 rods in 48 cases, and 5.5 mm and multi-rods in 43 cases, respectively. Visual analog scale (VAS) back, VAS leg, and ODI were significantly improved after surgery. Final VAS back and ODI were significantly higher in group P. There were no significant differences in the other items (Table 2).

### 3.2. Radiographic Parameters

PI-LL significantly improved from 37.3 ± 17.9° to 1.2 ± 12.2° after surgery. The other parameters also improved significantly after surgery (Table 3). In LL, an average of 15.5 ± 10.1° (4–42°) was obtained with PPS alone. After surgery, 60% of all cases had PI + 10 > LL > PI − 10, 70% had PI + 15 > LL > PI − 15, and 90% had PI + 20 > LL > PI − 20 (Figure 2). There was a low correlation between preoperative and postoperative LL and between preoperative PI-LL and the LL change using PPS. Meanwhile, a high correlation was observed between PI-LL after LLIF and postoperative PI-LL, and between preoperative PI-LL and total LL change. A moderate correlation was also observed between preoperative and postoperative PI-LL and between PI-LL after LLIF and LL change with PPS (Figure 3).

Comparing various spinopelvic parameters between the two groups, the pre- and postoperative PI, PI-LL after LLIF, pre- and postoperative PI-LL, and pre- and postoperative PT were significantly larger in group P (pre- and postoperative PI, *p* ≤ 0.001; pre- and postoperative PI-LL and PI-LL after LLIF, *p* ≤ 0.001; pre- and postoperative PT, *p* ≤ 0.001) than in group G. Postoperative LL was significantly smaller in group P (postoperative LL, *p* ≤ 0.001) than in group G (Table 4). Based on the results of univariate analysis, the four preoperative items (preoperative PI, preoperative PT, preoperative PI-LL, and PI-LL after LLIF) were independent variables, and postoperative PI-LL > 10° was the dependent variable. Logistic regression analysis revealed that PI-LL after LLIF and preoperative PI were risk factors for postoperative PI-LL > 10° (Table 5). ROC curve analysis of postoperative PI-LL > 10° using two items revealed that the cut-off values for each item were as follows: preoperative PI of 56° (area under the curve [AUC] 0.78) and PI-LL after LLIF of 20° (AUC 0.84) (Table 6).

### 3.3. Clinical Outcomes

VAS back, VAS leg, and ODI were significantly improved after surgery (*p* ≤ 0.001). The ODI improved from a preoperative average of 37.9 ± 6.2 to a final average of 22.1 ± 5.6 (*p* ≤ 0.001).

### 3.4. Complications

The rate of PJK was 13% (re-surgery, 8%), and there was no statistically significant difference in rod diameter and number. Rod fracture (RF) was observed in 20% of patients, and it occurred significantly more frequently when using two 5.5 mm rods than with two 6.0 mm rods and multiple 5.5 mm rods (5.5 mm vs. 6.0 mm, *p* = 0.008 and 5.5 mm vs. multi, *p* = 0.001). The rate of neurological deficits was 6% (three cases recovered within 3 months and two cases remained paralyzed). Transient thigh symptoms due to LLIF were observed in 39% of patients, but all patients recovered within 3 months. The proportions of patients with infection, breakage of SAI screw, and coronal imbalance after surgery were 1%, 4%, and 13%, respectively (Table 7).

## 4. Discussion

Improvements in various spinopelvic parameters and global alignment in ASD correlate with improvements in clinical outcomes and HRQOL [17]. Previous reports showed that traditional open ASD correction surgery results in a high percentage of complications and long-term hospitalization [17,18]. However, it has been reported that the CMIS in ASD is less invasive with reduced incidence of complications [14,19]. In addition, there are some reports on the usefulness of hybrid surgery with open posterior and Ponte osteotomy after LLIF [14,19]. However, lifting the intervertebral space with an LLIF cage also opens the facet joints and provides indirect decompressions of the spinal canal and intervertebral foramen. Subsequently, by applying rods, lordosis can be obtained in the process of closing the open facet joint. Therefore, even if it is a PPS procedure without Ponte osteotomy, sufficient correction is possible without causing nerve root impingement (Figure 4). Compared to the procedure of dissecting the posterior back muscles, exposing the lamina, and performing a Ponte osteotomy, it is obvious that correction surgery using LLIF and PPS in ASD is less invasive [14,19].

### 4.1. Pros and Cons of Staged Surgery

Staged surgery for adult spinal deformities has been reported. Tahn et al. reported that the incidence of complications between unstaged and staged surgery was not significantly different, but that staged surgery was associated with significant hospital stays and transfusions [20]. On the other hand, Anand et al. reported that staged surgery for ASD can reduce admission to the intensive care unit [21]. There are several reasons why CMIS involves two stages. The first is that the corrective impact of each surgery can be evaluated accurately. The second is to reduce the invasiveness of the surgery. As the duration of surgery increases, the amount of bleeding gradually increases, the blood is diluted and, as a result, hemostatic functions deteriorate, and the amount of bleeding is anticipated to increase. The third reason is that the surgical method can be reconsidered by evaluating the alignment and changes in neurological symptoms after the first surgery. Anand et al. reported that after performing a multi-level LLIF, the alignment was evaluated in a standing position to determine whether to use the PPS procedure or the open procedure in the second surgery. If sufficient correction is achieved in the first surgery, the PPS procedure is indicated, or the fixed range may be reduced in some cases. If sufficient intervertebral release and correction are not achieved after the first surgery, open surgery might be needed. In addition, if indirect decompression by LLIF does not produce sufficient improvements in neurological symptoms, direct decompression must be performed in the second surgery. Two-stage surgery is very useful because such evaluation and reconsideration of the surgical procedure can be performed. The only disadvantage of staged surgery is the extended duration of hospital stay. However, this is not a big problem considering the above-mentioned advantages.

### 4.2. Correction Force/Indications for CMIS for ASD

Reports of CMIS for ASD are gradually increasing, but there are many reports on its ineffectiveness in cases of severe deformation. Mummaneni et al. reported the minimally invasive spinal deformity surgery (MISDEF) algorithm based on the severity of deformity and spinopelvic parameters [22]. They suggested that patients with mild deformities (PI-LL < 30° and PT < 25°) can undergo CMIS. Furthermore, they indicated MISDEF2 as multi-level MIS in patients with PI-LL < 30° and CMIS with ACR or mini-open PSO in patients with PI-LL ≥ 30° [23]. Haque et al. reported that the amount of postoperative SVA change was larger in conventional open surgery than in CMIS, and postoperative PI-LL was larger in CMIS than in hybrid and conventional open surgery [14]. This suggests that the corrective force in the MIS procedure is weaker than that in the traditional posterior open surgery. Park et al. and Mundis et al. reported that ASD patients who underwent combined severe fixed sagittal imbalance and spinopelvic malalignment were poor candidates for MIS surgery alone due to the high risk of residual postoperative deformity and fixed sagittal imbalance [19,24]. In this study, the following were noted: a low correlation between pre- and postoperative LL and between preoperative PI-LL and LL change with PPS, a high correlation between preoperative PI-LL and total LL change and between post-LLIF PI-LL and postoperative PI-LL, and a moderate correlation between pre- and postoperative PI-LL and between post-LLIF after PI-LL and LL change with PPS. This indicates that higher correction is achieved in cases with severe spinal deformity and that PI-LL after LLIF strongly affects postoperative PI-LL. This is similarly suggested by the results of the multivariate analysis. Furthermore, the preoperative mean PI-LL was 35° and the preoperative mean PT was 31° in this study, indicating that the degree of deformity in patients with ASD was higher compared to those in previous reports on CMIS for ASD [14,15]. This suggests that sufficient anterior intervertebral release by LLIF and the creation of an appropriate rod contour can achieve sufficient correction for severe spinal deformity with posterior fixation with PPS. In this study, logistic regression analysis revealed that PI-LL after LLIF and preoperative PI were risk factors of postoperative PI-LL > 10°, and ROC analysis of postoperative PI-LL > 10° using two items revealed that the cut-off values for each item were as follows: preoperative PI of 56° and PI-LL after LLIF of 20°. This suggested that the tolerance for target LL is greater in patients with high PI, indicating that this is similar to the previous formula [25,26]. Since the flexibility of spinal deformity changes remarkably after LLIF, it is difficult to use preoperative parameters as the indication criteria for CMIS. From the results of ROC analysis, we believe that PI-LL after LLIF ≤ 20° can be used as one of the indication criteria for CMIS.

### 4.3. Complications

Park et al. reported that the hybrid group demonstrated higher absolute improvement in radiographic parameters at the expense of a higher complication rate compared with the CMIS group. The complication rate (major or minor) in the hybrid group was 55% compared to 33% in the CMIS group [19]. This finding correlates with that of Wang et al., who demonstrated a major complication rate of 40% in patients undergoing hybrid surgery compared to 14% in the CMIS group [27]. Complications specific to the LLIF approach include transient anterior thigh dysesthesias and hip flexor weakness, which occur in 15% to 40% of patients undergoing LLIF [28,29,30]. In this study, the incidence of thigh symptoms was 39% in patients who underwent LLIF, which was almost similar to that reported in previous reports. However, all the patients recovered within 3 months, which was not clinically significant.

### 4.4. Proximal Junctional Kyphosis

Proximal junctional kyphosis (PJK) and proximal junctional failure (PJF) are common potential postoperative complications [31]. There are various reports on the incidence of PJK after correction surgery for ASD, which ranges from approximately 20–50% [11,31,32]. PJK is considered to be caused by multiple factors, including age, thoracolumbar spine muscle mass, overcorrection, level of UIV, preoperative SVA, fixation to ilium, and terminal rod contour [33,34,35]. Rhee et al. reported that PJK is seen less frequently in anterior spinal fusion than in posterior spinal fusion [36]. Some studies report that the reduction in posterior soft tissue damage using PPS is useful in the prevention of PJK [13,34,37]. In this study, the incidence of PJK was approximately 13% and the re-surgery rate was 8%, which is lower than that in previous reports [32,33,34]. The first reason for the low incidence of PJK is due to the effect of a reduction in the posterior soft tissue damage by using PPS, which is similar to those of past reports [13,34,37]. The second reason is ensuring sufficient kyphosis of terminal rod contour, suitable for postoperative reciprocal change, as in our previous report [35]. Moreover, in this study, the incidence of PJK tended to be higher when a 6 mm rod was used than when a 5.5 mm rod was used. As reported by Cahill and Lange, the use of softer implants is predicted to reduce the load on the junction, reduce adjacent disc degeneration, and reduce the occurrence of PJK [34,38]. A transition rod that has high rigidity at the lumbar level and low rigidity at the thoracic spine may be useful for preventing PJK in the future, and development is desired.

### 4.5. Rod Fractures

RF is a frequent implant-related complication following ASD surgery with an incidence of 6.8–22% [39,40]. In this study, the incidence of RF was 21%. The incidence of RF was 30% in patients who received the 5.5 mm rod, which is higher than that in previous reports, but the frequency is equal or less than that in those who received the 6 mm and multi-rod. This is because in CMIS, the paraspinal muscles were not detached, the spinal lamina was not exposed, and bone grafting was not performed [15]. In addition, since posterior facet joint resection was not performed, the facet joint worked as a hinge when LL was formed with the ideal rod. As a result, there were cases where the contact between the vertebral body and cage was insufficient. We considered this to be the reason for slow bone fusion and higher incidence of RF in cases using 5.5 mm rod than in conventional open surgery. Moreover, RF occurred more frequently with the usage of two 5.5 mm rods than with two 6.0 mm rods and 5.5 mm multi-rods in this study. Merrill et al. reported the usefulness of multi-rod application as a countermeasure for RF [41]. In this study, the usefulness of 6 mm rods and multi-rods in the prevention of RF was clear. It is possible to increase the bone fusion rate and reduce RF by increasing the rod diameter and the number of rods, increasing the durability of rods, and delaying the time of breakage due to wear, in CMIS procedures that allow time for bone fusion. Izeki et al. reported that spontaneous facet fusion was confirmed in patients with degenerative disease who underwent surgery with LLIF and PPS [42]. He noted that spontaneous fusion may be achieved in patients with facet degeneration. In ASD patients who underwent CMIS, both anterior intervertebral bone fusion by LLIF and spontaneous facet fusion without bone grafting will be achieved over time.

### 4.6. Infection

Haddad et al. reported that the incidence of deep wound infection was 16% in traditional open surgery [43]. Eastlack et al. and Uribe et al. reported that postoperative infections following ASD is lower in MIS than in hybrids [15,44]. Eastlack et al. reported that the postoperative infection rate in ASD was 1.5% [14], and it was 1% in this study, which is similar. There are two reasons for the low infection rate of CMIS in ASD. The first reason is the reduction in paraspinal muscle detachment due to the use of PPS and the consequent reduction in exposure to bacteria. The second one is the reduction in operative time and blood loss. The operative time was shorter and blood loss was lesser in CMIS than in hybrid surgery with LLIF and posterior open instrumentation [19,45].

We have shown that CMIS using LLIF and PPS is indicated for de novo scoliosis, spinal deformity with vertebral body fracture, and adult idiopathic scoliosis (AS) without interbody fusion or only one level of intervertebral fusion. Contraindications are cases of iatrogenic kyphosis with multi-level vertebral fusion and posterior element bone fusion and cases of AS with multi-vertebral bone fusion (Table 1). However, it is important to reassess the flexibility of the spine after performing LLIF. Since there are many cases in which flexibility is significantly changed due to the release of anterior interbody bridging and posterior facet joints due to LLIF, reassessment of flexibility after LLIF is important.

### 4.7. Limitations

This study has several limitations. First, the sample size was small. Additional studies with large sample size are required for validating the results. The second limitation is related to the lack of bone mineral density (BMD) evaluation. In this study, the average age of the patients was 73.3 years, and 88% of the total patients were women; consequently, most of them would have an osteoporotic spine. Low vertebral bone density alters the biomechanical impact of implants on vertebral bodies and may increase implant-related complications, such as vertebral fractures and implant failure. Thus, we believe that a large-scale prospective study including the evaluation of BMD will be necessary in future. The third limitation is that the follow-up period differed, depending on the rod diameter and material. Since the patients who received multiple rods in this study had the lowest follow-up period, RF may increase in the future. Thus, further long-term follow-up is important. The fourth limitation is that the target alignment in this study was PI-LL < 10°. There are some reports that the target alignment differs between patients with low PI and those with high PI [25,26]. In addition, the ideal alignment differs depending on the age [26]. Further research on the achievement conditions for target alignment according to age and PI is desired in the future.

## 5. Conclusions

We reported the clinical results of CMIS in ASD and clarified the conditions for achieving postoperative PI-LL < 10°. Sufficient correction was obtained using CMIS with LIF and PPS. It was suggested that if PI-LL is ≤20° after LLIF, postoperative PI-LL ≤ 10° can be achieved even with correction by PPS.

## Figures and Tables

**Figure 1 jcm-11-01586-f001:**
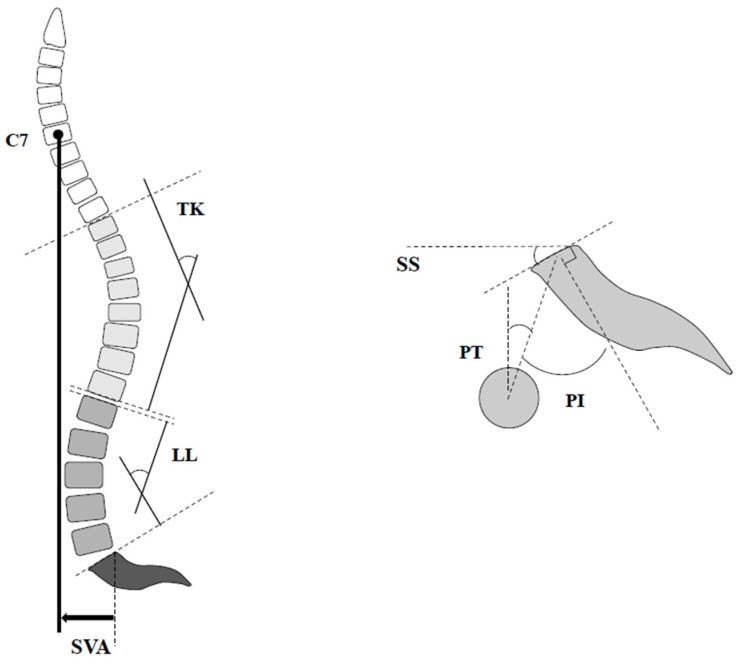
Illustration of various spinopelvic parameters. LL: lumbar lordosis; TK: thoracic kyphosis; PI: pelvic incidence; PT: pelvic tilt; SS: sacral slope; SVA: sagittal vertical axis.

**Figure 2 jcm-11-01586-f002:**
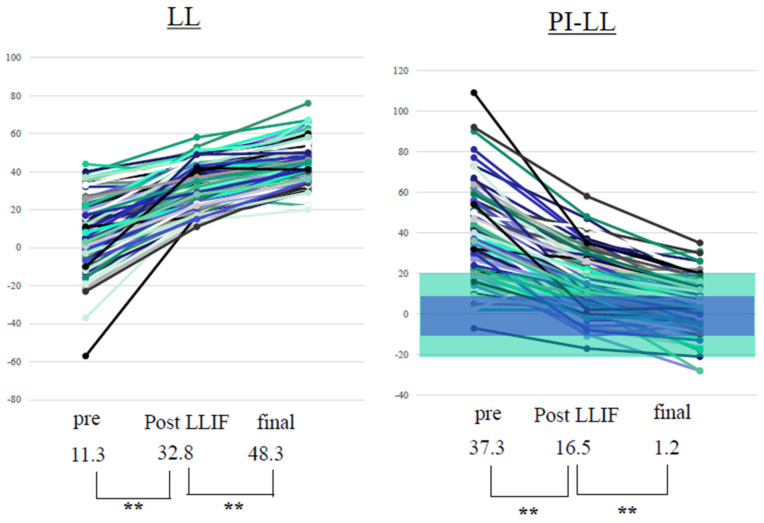
Preoperative and postoperative LL and PI-LL. Values are presented as mean ± standard deviation. Wilcoxon signed-rank test. ** Statistically significant. PPS improved LL by 15.5° (4–42°). After CMIS, 60% of all patients met PI + 10° ≥ LL ≤ PI − 10° (blue range) and 90% met PI + 20 > LL > PI − 20 (green range). LL: lumbar lordosis; PI: pelvic incidence; PPS: percutaneous pedicle screw; LLIF: lateral lumbar interbody fusion. ** *p* < 0.001.

**Figure 3 jcm-11-01586-f003:**
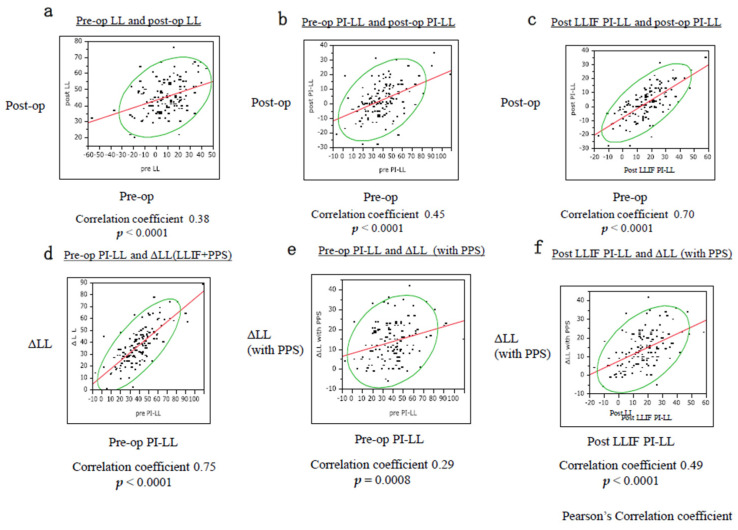
Correlation between various parameters. There was a low correlation between preoperative and postoperative LL (**a**) and between preoperative PI-LL and LL change using PPS (**e**). A high correlation was observed between PI-LL after LLIF and postoperative PI-LL (**c**) and between preoperative PI-LL and total LL change (**d**). A moderate correlation was observed between preoperative and postoperative PI-LL (**b**) and between PI-LL after LLIF and LL change with PPS (**f**). These findings indicate that greater correction is achieved in cases of greater deformity. LL: lumbar lordosis; ΔLL: change in LL; PI: pelvic incidence; LLIF: lateral lumbar interbody fusion; PPS: percutaneous pedicle screw.

**Figure 4 jcm-11-01586-f004:**
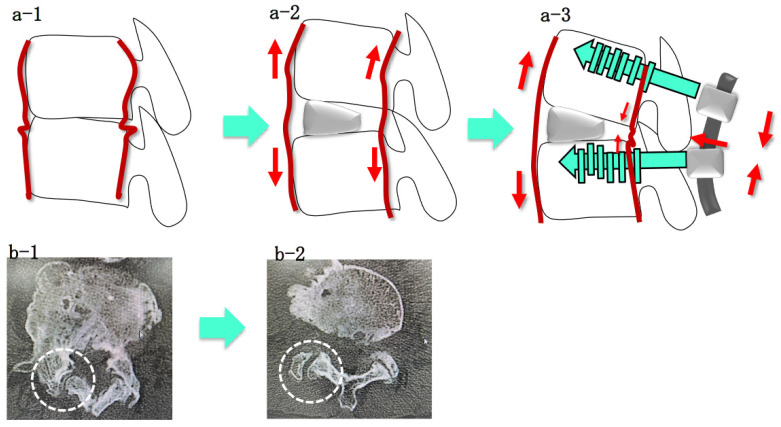
The mechanism of correction surgery using LLIF and PPS. Spreading the intervertebral space with insertion of a LLIF cage also opens the facet joints and provides indirect decompression of the intervertebral foramen. Subsequently, by applying a rod, lordosis can be obtained by closing the facet joint. Therefore, even if PPS procedure without Ponte osteotomy was undertaken, sufficient correction is possible without causing nerve root impingement (**a-1**,**a-2**,**a-3**). The contracture facet joints were released and dilated with insertion of LLIF. LLIF procedure can dissociate not only the anterior element, but also the posterior element indirectly (**b-1**,**b-2**). LLIF: lateral lumbar interbody fusion; PPS: percutaneous pedicle screw.

**Table 1 jcm-11-01586-t001:** Indications for CMIS in ASD.

Spinal Disease		Indication
De novo kyphoscoliosis		GI
Spinal deformity with OVF		GI (with corpectomy)
Degenerative kyphosis		UI
Adult scoliosis	without bone union	UI
	anterior bone union	UI
	anterior and posterior bone union	UNI (with mini-open Ponte)
Iatrogenic kyphosis	one level	UI
	Multi-level	NI

Cases with posterior elemental bone fusion and multi-level iatrogenic kyphosis are not indicated for CMIS. CMIS: circumferential minimally invasive surgery; ASD: adult spinal deformity; OVF: osteoporotic vertebral fractures. GI: Good indication; UI: Usually an indication, but sometimes not; UNI: Usually not an indication, but can be an indication for mini-open Ponte osteotomy; NI: Not an indication.

**Table 2 jcm-11-01586-t002:** Demographic data.

Parameter		Whole Group (*n* = 145)	Group G (*n* = 110)	Group P (*n* = 35)
Age (years)		73.3 ± 6.5 (48–83)	73.3 ± 6.9	73.0 ± 7.7
Rate of women (%)		88	80.8	78.1
Period of follow-up (months)		40.7 ± 6.2 (30–54)	40.7 ± 6.3	40.6 ± 6.2
Rod diameter/number in construct	5.5 mm/2 rods	54 (61%)	40	14
	6 mm/2 rods	48 (30%)	33	15
	5.5 mm/3 rods	43 (1%)	33	10
Number of levels fused		10.3 ± 0.5 (10–13)	10.3 ± 0.5	10.4 ± 05
Number of LLIF		4.0 ± 0.5 (3–6)	4.0 ± 03	4.1 ± 05
Number of patients with corpectomy(case)		8	7	1
UIV (case)	T7	3 (1%)	3	0
	T9	48 (29%)	34	14
	T10	94 (70%)	72	22
Operative time (min)	Anterior (first surgery)	109.6 ± 37.5 (59–273)	112.4 ± 40.2	100.8 ± 26.5
	Posterior (second surgery)	233.3 ± 50.9 ** (171–290)	233.2 ± 52.1 **	233.0 ± 47.7 **
Blood loss (mL)	Anterior (first surgery)	104.3 ± 139.3 (0–970)	117.7 ± 152.4	63.0 ± 75.1
	Posterior (second surgery)	498.2 ± 305.7 ** (79–1530)	488.6 ± 290.9 **	528.1 ± 350.9 **
VAS back	Before surgery	6.6 ± 1.4	6.4 ± 1.3	7.2 ± 1.2
	Final	2.8 ± 0.8 *	2.6 ± 0.7 *#	3.3 ± 0.9 *
VAS leg	Before surgery	5.4 ± 2.2	5.5 ± 2.1	5.0 ± 2.6
	Final	1.7 ± 1.0 *	1.7 ± 1.1 *	1.7 ± 1.1 *
ODI	Before surgery	37.9 ± 6.2	37.3 ± 6.2	40.1 ± 5.1
	Final	22.1 ± 5.6 *	20.8 ± 5.8 *#	25.9 ± 2.6 *

Values are presented as mean ± standard deviation. Wilcoxon signed-rank test. * Statistically significant. UIV: upper instrumented vertebra; LLIF: lateral lumbar interbody fusion. VAS: visual analog scale; ODI: Oswestry Disability Index. * *p* < 0.001 compared with before surgery, ** *p* < 0.001 compared with first surgery, # *p* < 0.001 compared with group P.

**Table 3 jcm-11-01586-t003:** Spinopelvic parameters.

Parameter	Pre-Op	Post-Op	Final	*p* Value (Pre-Op vs. Final)
PI	47.3 ± 10.5	48.0 ± 11.1	47.9 ± 10.8	0.542
PI-LL	37.3 ± 17.9	1.2 ± 12.2	2.7 ± 12.1	<0.001 *
LL	11.3 ± 15.9	48.3 ± 10.3	46.5 ± 10.8	<0.001 *
PT	31.8 ± 11.2	17.5 ± 9.8	18.5 ± 9.2	<0.001 *
TK	19.4 ± 16.3	39.2 ± 10.8	42.3 ± 11.6	<0.001 *
SVA	83.3 ± 50.1	15.7 ± 35.0	38.0 ± 36.5	<0.001 *

Values are presented as mean ± standard deviation. Wilcoxon signed-rank test. * Statistically significant. There was significant improvement in all parameters, except PI, after surgery. LL: lumbar lordosis; PI: pelvic incidence; TK: thoracic kyphosis; PT: pelvic tilt; SVA: sagittal vertical axis.

**Table 4 jcm-11-01586-t004:** Spinopelvic parameters (group P vs. group G).

Variable		Group P (*n* = 35)	Group G (*n* = 110)	*p* Value
PI	Pre-PI	57.1 ± 10.6	44.8 ± 10.4	<0.001 *
	Post-PI	58.2 ± 10.7	46.5 ± 10.7	<0.001 *
LL	Pre-LL	10.4 ± 13.8	11.5 ± 16.7	0.590
	LL after LLIF	28.5 ± 10.4	33.3 ± 11.1	0.065
	Post-LL	41.4 ± 10.0	50.4 ± 9.5	<0.001 *
ΔLL	ΔLL (with LLIF)	21.1 ± 11.2	21.7 ± 12.9	0.776
	ΔLL (with PPS)	10.0 ± 8.8	17.1 ± 10.0	0.109
	Total ΔLL	31.1 ± 15.1	38.8 ± 16.7	0.210
PI-LL	Pre-PI-LL	44.6 ± 17.1	35.0 ± 17.7	<0.001 *
	PI-LL after LLIF	26.7 ± 7.7	13.1 ± 12.0	<0.001 *
	Post-PI-LL	16.7 ± 5.2	−3.8 ± 8.8	<0.001 *
PT	Pre-PT	37.6 ± 11.5	29.9 ± 10.6	<0.001 *
	Post-PT	26.3 ± 7.6	14.8 ± 8.5	<0.001 *
TK	Pre-TK	18.9 ± 15.2	19.6 ± 16.9	0.305
	Post-TK	36.4 ± 8.8	37.5 ± 11.4	0.625

Values are presented as mean ± standard deviation. Mann–Whitney U test. * Statistically significant. LLIF: lateral lumbar interbody fusion; LL: lumbar lordosis; PI: pelvic incidence; TK: thoracic kyphosis; PT: pelvic tilt.

**Table 5 jcm-11-01586-t005:** Postoperative PI-LL > 10° risk factors (logistic regression analysis).

	Odd Ratio	95% CI	*p* Value
Pre-op PI	0.93	0.87–0.99	0.032 *
PI-LL after LLIF	0.87	0.79–0.95	0.001 *
Pre-op PT	0.98	0.90–1.06	0.618
Pre-op PI-LL	1.04	0.99–1.10	0.068

* Statistically significant. LLIF: lateral lumbar interbody fusion; LL: lumbar lordosis; PI: pelvic incidence; PT: pelvic tilt; CI: confidence interval.

**Table 6 jcm-11-01586-t006:** ROC curve analysis.

	Cut-Off Value	Sensitivity	Specificity	AUC
PI-LL after LLIF	20°	0.95	0.65	0.846
PI	56°	0.75	0.83	0.781

On performing an ROC analysis of postoperative PI-LL > 10° using two items, the cut-off values for each item were as follows: PI-LL after LLIF of 20° (AUC 0.84) and PI of 56° (AUC 0.78). ROC: receiver operating characteristic; LLIF: lateral lumbar interbody fusion; LL: lumbar lordosis; PI: pelvic incidence; AUC: area under the curve.

**Table 7 jcm-11-01586-t007:** Complications.

*n* = 145			≤30 Days	30 Days 3 Years	After 3 Years	Total	*p* Value
PJK	5.5 mm, 2 rods (*n* = 54) (case)		3	4	0	7 (13%)	n.s.
	6 mm, 2 rods (*n* = 48) (case)		2	6	0	8 (17%)	
	5.5 mm, multi-rods (*n* = 43) (case)		1	3	0	4 (9%)	
	**Total (case)**		**6**	**13**	**0**	**19 (13** **%)**	
	Revision (case)		3	9	0	12 (8%)	
Rod fracture	Rod diameter, number	5.5 mm, 2 rods (*n* = 54) (case)	0	18	2	20 (37%)	5.5 mm vs. 6 mm: 0.008 * 5.5 mm vs. multi: 0.001 *
		6 mm, 2 rods (*n* = 48) (case)	0	7	0	7 (15%)	
		5.5 mm, multi-rods (*n* = 43) (case)	0	4	0	4 (9%)	
		**Total (case)**	**0**	**29**	**2**	**31 (21%)**	
	Reasons	Nonunion (case)	0	19	0	19 (13%)	
		ALL rupture (case)	0	9	0	9 (6%)	
		After union (case)	0	1	2	3 (2%)	
	Revision (case)		0	28	0	28 (19%)	
Neurological deficit (case)			5 (transient 3, permanent 2)	0	0	5 (3%)	
Thigh symptom (case)			56 (transient)	0	0	56 (39%)	
Infection (case)			1	1	0	2 (1%)	
Breakage of SAI (case)			0	6 (revision 3)	0	6 (4%)	
Coronal imbalance (case)			16	2	1	19 (13%)	

Chi-square test. * Statistically significant. PJK: proximal junctional kyphosis; ALL: anterior longitudinal ligament; SAI: S2 alar iliac screw; n.s.: not significant.

## Data Availability

The data used in this study are available upon reasonable request from the corresponding author.

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
