# Peer review of "Conditions for Achieving Postoperative Pelvic Incidence-Lumbar Lordosis < 10° in Circumferential Minimally Invasive Surgery for Adult Spinal Deformity"

_jcm, 2022, doi:10.3390/jcm11061586_

Round 1

Reviewer 1 Report

The manuscript “Conditions for achieving postoperative pelvic incidence-lumbar lordosis <10° in circumferential minimally invasive surgery for adult spinal deformity” by Masayuki Ishihara, Shinichirou Taniguchi, Takashi Adachi, Yoichi Tani, Masaaki Paku, Muneharu Ando, Takanori Saito aimed to evaluate the clinical outcomes of circumferential mini- mally invasive surgery (CMIS) using lateral lumbar interbody fusion (LLIF) and percutaneous pedicle screw (PPS) in adult spinal deformity (ASD) patients, and to clarify the conditions for achieving postoperative pelvic incidence-lumbar lordosis (PI-LL)<10°.

Below are my comments and remarks regarding the article:

1. Introduction requires expansion. The subject matter relates only to the surgical procedure and does not describe information on the sagittal balance and does not include the purpose of the study.
2. The information in the materials and methods "This study was approved ..." can be moved to the end of this chapter.
3. Lack of sample x-rays before and after intraoperative ...
4. Table 1. Instead of symbols, please use text or text abbreviations - it will improve legibility.
5. Table 2. Demographic data does not present only the clinical characteristics of the studied group.
6. I did not find the results of the correlation of the sagittal balance parameters with the ODI, although it relates to the beginning of the discussion

Author Response

We thank you for taking the time and effort necessary to review our manuscript and provide us with these valuable comments and suggestions. Accordingly, we revised our manuscript and made changes to it. Please note that the yellow highlighted parts represent our responses to the comments.

Reviewer 2 Report

The authors present the radiographic parameters after MIS adult deformity correction. The paper has 2 major flaws:

  1. The authors report the results of a single surgery while they actually performed 2 separate surgeries a week apart. surgical time and EBL cant be presented as a sum as this is misleading information.
  2. The report on clinical results is minimal with no elaboration regarding neurlogical outcome. 

Author Response

(The authors gave the same response as above.)

Reviewer 3 Report

Thank you for the opportunity to review this article. The article retrospectively analyses a series of patients with adult spinal deformities in order to evaluate the results in terms of sagittal balance of circumferential minimally invasive surgery with LLIF and PPS. 

The article is interesting and provides novel data that can contribute to the literature on the topic. In my opinion the article is suitable for publication, although there are a few points that need to be clarified.

1) In Introduction you said: "Schwab et al. reported that the Oswestry Disability Index (ODI) was significantly lower in patients with sagittal alignment achieving postoperative pelvic tilt [PT]<20°, sagittal vertical axis [SVA]<100mm, and postoperative pelvic incidence-lumbar lordosis (PI-LL)<10°[7]", by citing: "Schwab F, Patel A, Ungar B, Farcy JP, Lafage V, et al. Adult spinal deformity-postoperative standing imbalance: Adult spinal deformity-postoperative standing imbalance: how much can you tolerate? An overview of key parameters in assessing alignment and planning corrective surgery. Spine (Phila Pa 1976). 2010 Dec 1;35(25):2224-31. doi: 10.1097/BRS. 0b013e 3181ee6bd4.PMID: 21102297 Review." This article is a milestone on the topic. However, Schwab et al., unlike you stated, reported the difference in outcome using the Oswestry Disability Index between patients with SVA<50mm or SVA>50mm and between patients with PI-LL<9° or PI-LL>9°. Please be more precise in your citation and justify in the introduction or better in the methods section why you used different, though in my opinion legitimate, thresholds to stratify patients.

2) Please explain better in the discussion section the rationale of performing LLIF and PPS in two different stages. Reading the article in this form, it seems that the only advantage is to be able to differentiate the individual contribution of the two techniques to the correction. 

3) The description of the statistical analysis in the methods section is too poor and needs to be improved. The descriptions of multivariate analysis and ROC analysis are completely missing, although both are present in the results. Moreover, it seems to me extremely restrictive to limit the univariate analysis to Student's t-test and Fisher's exact test. In the case of the T-test, the use of a parametric test must be justified by the characteristics of the sample through preliminary analyses of normality and homogeneity of variances. In the case of the Fisher test, I believe that the large sample size analysed in this study would suggest that other chi-square tests are more appropriate. I have no doubt that the significance of the results obtained would not be dramatically altered even by changing the approach to the univariate analysis, but I would like a more solid evaluation to be made, or the choices made to be justified.

Thank you.

Author Response

(The authors gave the same response as above.)

Round 2

Reviewer 1 Report

No more comments. I leave the decision to the Editor

Author Response

(The authors gave the same response as above.)

Reviewer 2 Report

Review according to the previous comments was partially performed.

Author Response

(The authors gave the same response as above.)

Reviewer 3 Report

Thank you. The authors have addressed all my concerns. I consider this article suitable for publication in its current form.